# Challenges for Optimum Cardiopulmonary Resuscitation in the Emergency Departments of Limpopo Province: A Qualitative Study

**DOI:** 10.3390/healthcare11020158

**Published:** 2023-01-04

**Authors:** Livhuwani Muthelo, Hendrica Mosima Seimela, Masenyani Oupa Mbombi, Rambelani Malema, Arthur Phukubye, Lerato Tladi

**Affiliations:** 1Department of Nursing Science, University of Limpopo, Sovenga St, Polokwane 0727, South Africa; 2Department of Physiology and Environmental Health, University of Limpopo, Sovenga St, Polokwane 0727, South Africa

**Keywords:** challenges resuscitation team, professional nurses, medical doctors

## Abstract

Aim: To describe the challenges for optimum resuscitation processes in Emergency Departments in Limpopo Province, South Africa. Design: A qualitative explorative research approach was adopted to explore the resuscitation team’s experiences in Emergency Departments. Method: Five medical doctors and twelve professional nurses were purposively sampled to participate in the study. The depth of the information obtained from the participants determined the sample size. Data collected from semi-structured individual interviews were analyzed using thematic analysis. Data quality was ensured by applying four elements: credibility, transferability, dependability, and confirmability. Results: The study findings indicated diverse challenges for optimum resuscitation processes that include: A general shortage of emergency personnel, the lack of material resources and the unavailability of funds for payment of national and international trauma symposiums, the poor maintenance of emergency equipment, the lack of a continuous training program and the resuscitation team receiving different instructions from various team leaders about the standardized procedures and policies of the resuscitation process. The team leaders and managers often blamed, depreciated and disregarded the resuscitation team for failed resuscitation efforts. Public contribution: The study findings are a point of reference for the emergency resuscitation team and the department of health policymakers. Trained and well-equipped emergency resuscitation teams can improve the quality of life for patients with cardiac arrest.

## 1. Introduction

Emergency Departments (EDs) are at the front line in the hospitals with a critical role in ensuring access to and efficient care of acute illness and injuries in the healthcare system. However, the EDs environment is burdened with complex patient loads [1,2], long shifts, and administrative challenges resulting in poor patient care, high pressure, and high-volume workloads among the staff members [3]. Salway, Valenzuela, Shoenberger et al. [2] noted that overcrowding and the associated impacts, like increased waiting times, ambulance availability, diversion, length of stay, medical errors, patient mortality, and hospital harm due to financial losses in ED all act as a burden to the resuscitation team. The staff members in the EDs deal with unpredictable events and life-threatening medical emergencies such as Tension Pneumothorax, Cardiac Tamponade, and Status Asthmaticus. Some of these conditions complicate Cardiac Arrest which has an unknown outcome. In this regard, Castelao, Boos, Ringer et al. [4] noted that a well-coordinated and executed Cardiopulmonary Resuscitation (CPR) contributes to a better Cardiac Arrest outcome. Thus, effective CPR requires a psychologically healthy, skilled, and composed resuscitation team. Nevertheless, the stress level brought on by the workload burden is likely to lead to more mistakes by the ED staff leading to an unsuccessful CPR and providing an opportunity for medico-legal claims.

Cardiopulmonary resuscitation combines chest compressions and rescue breathing as the basis for Basic Life Support (BLS) [5]. Various authors have noted that South African healthcare settings have a different number of healthcare professionals trained and untrained on BLS who are responsible for the provision of resuscitative care [6,7]. Furthermore, errors during resuscitation might result from nurses’ and doctors’ lack of training and insufficient knowledge and skills regarding triage.

Abraham, Thom, Greenslade et al. [8] noted global trends recognizing EDs as stressful environments for resuscitation teams. The authors, as mentioned earlier, further indicated that the stressful EDs were related to the increasingly high workload, moderate self-realization, low levels of conflict and nervousness, poor skill mix, and overcrowding which influenced the perceptions of the resuscitation team about the emergency environment. In the United States (USA), EDs are more stressful and busier, with a patient workload of more than 131 million total visits to the ED in 2011 [9]. While in Sweden, Källberg, Ehrenberg, Florin et al. [10] reported that the resuscitation team is confronted with administrative challenges, such as a lack of organization and control in the EDs with a high workload that affects the provision of quality patient care. In addition, burnout amongst physicians was also found to contribute to poor quality care, increased medical errors, and attrition from medical practice exacerbating the shortage and maldistribution of emergency personnel [11].

Many countries in Sub-Saharan Africa face the challenge of having an integrated emergency care system as recommended by the World Health Assembly due to the diverse burden of acute diseases experienced in the hospitals [12]. Reynolds, Mfinanga, Sawe et al. [12] noted that many African countries face challenges such as a lack of capacity for ED personnel as well as infrastructure and resources-related challenges. Thus, the University of Botswana’s resuscitation training project has developed a strategy to support the emergency resuscitation teams. However, the university experienced challenges with the project implementation due to difficulties in staff retention, maintenance of educational courses, and ongoing financial support [13].

South Africa is no exception to other African countries confronted with the challenges of providing quality emergency care. The patient waiting period has been deliberated several times by different authors in South Africa who concurred that a triage system would be the best option to reduce the problem [14,15,16]. A study conducted at George Mukhari Hospital ED reported many medical doctors and nurses blamed themselves and felt incompetent and helpless when the patients failed to survive during resuscitation. As a result, some of these health professionals resorted to working longer hours with the hope of restoring the loss, thus suggesting the element of denial. The feeling of denial that manifests as one rejects the emotional feeling often results in chronic grief and depression, negatively affecting the resuscitation team in the EDs [17]. Only after numerous failed resuscitation attempts will the resuscitation team call it quits. This terrible encounter takes a physical and emotional toll on the team, who is the patient’s last hope of survival. The death of a patient causes enormous personal and professional stress and anxiety for the resuscitation team [18] The above study indicates that there is a need for intervention measures to assist the resuscitation team in EDs in overcoming the emotional reactions experienced during the resuscitation process. Based on the above background, the resuscitation team are subject to different demands in the EDs, which impact the provision of quality emergency care services; yet, little scholarly work has been conducted around the area. Understanding the obstacles to optimum resuscitation promises a significant outcome for the reported appalling medico-legal hazards around Limpopo Province and improving the patient safety culture [19] This can also stand as a baseline reference for enhancing quality emergency care with fewer errors, reducing delivery delays, improving efficiency, increased market share, and lower cost [20].

## 2. Methodology

The study adopted a qualitative explorative research design to explore the resuscitation team’s challenges. The qualitative research approach emphasizes the description of human experiences and insists on carefully portraying everyday life as people experience it [21]. The exploratory design explored the dimensions of the resuscitation team’s challenges in EDs [22].

### 2.1. Study Setting

The study was conducted in the emergency departments of two tertiary hospitals in the level three category. Hospital A has twenty-two wards, fourteen specialized outpatient clinics, two resuscitation units (trauma and medical/surgical), and one triage area. The resuscitation team within hospital A consisted of 11 doctors and 21 professional nurses. Hospital B has fourteen wards and eleven specialized clinics of the Outpatient Department. Its EDs have 32 Professional nurses and 12 Medical doctors who constitute the resuscitation team. Both hospitals serve as referral centres for all complicated emergency medical conditions for all the districts and regional hospitals of the Limpopo Province.

### 2.2. Population and Sampling

The resuscitation team consisted of all the professional nurses and medical doctors working in the EDs of the selected two tertiary hospitals in South Africa. Purposive convenience sampling was used to choose the available resuscitation team members in both EDs. The resuscitation team members were the best participants for the study due to their knowledge of and qualification level with resuscitation. For instance, the selected participants had practical experience in resuscitation as well as a diploma in comprehensive trauma nursing care and MBChB, making them suitable candidates for the resuscitation team. The community service nurses in the nursing category were excluded as they are still under supervision, as well as the intern medical officers. Any resuscitation team member with less than two years of working experience was excluded from the study. Twenty-four professional nurses and eleven medical doctors were purposively selected for the study.

### 2.3. Data Collection

Experienced, trained researchers conducted semi-structured individual interviews with an interview guide in the selected counselling rooms of the EDs. According to Galletta and Cross [23] using a semi-structured interview in a qualitative method combines a pre-determined set of open questions with the opportunity for the interviewer to explore a particular theme or response further. The resuscitation team was recruited to participate in the study with the assistance of the ED nurse manager. To build rapport with the resuscitation team, the authors introduced themselves and explained the benefits of participating in the study, the study’s aim, and that participation in the study was voluntary. The interviews were recorded using a digital voice recorder, and field notes were written to collect data from the participants as they responded. One central question, “Will you please describe the challenges you experience with resuscitation?” was asked of each participant. The interview guide was written in English since the professional nurses, and medical officers were conversant in English. Probing questions were used after the first response to gather more information about their experiences. Probing questions usually seek more information about a particular topic and encourage the person to elaborate on the already provided information [24]. Each interview lasted approximately 15 to 30 min to gain an in-depth understanding of the resuscitation team’s challenges in the selected emergency units. The study was conducted within the hospital premises where COVID-19 safety regulations were already implemented, including social distancing, sanitizing after each participant, and wearing masks.

### 2.4. Data Analysis

The thematic analysis approach was utilized to identify, examine, and report patterns/themes within the qualitative data gathered [25]. All audio-recorded interviews were captured verbatim in a Microsoft-word document. Four primary authors carefully read through the transcriptions of each interview to get a sense of the research as a whole. Interview transcripts and field notes were analyzed to generate a list of similar topics that were clustered together and formed into columns. Related topics were grouped into categories. The primary author assembled the data belonging to each category together, and a preliminary data analysis was performed to generate the theme and sub-themes. After the co-authors listened to the recorded interviews and the transcribed data, the study findings were discussed, and the final theme and sub-themes were identified and summarized. There were fewer inconsistencies in the identified themes and sub-themes, which led to other themes collapsing into one theme. The remaining four authors verified the identified theme and sub-themes and a consensus was reached by all the authors.

## 3. Results

### 3.1. Theme 1: Diverse Challenges for Optimum Resuscitation Processes

The study findings indicate diverse challenges for optimum resuscitation processes at the two selected tertiary hospitals. These challenges include a shortage of human and material resources for emergency care services. The resuscitation team expected a training and education program to improve their skills and experiences in emergency care which was not provided due to a lack of available funds for payment for national and international trauma symposium attendance. This resulted in the lack of a continuous training program that impacted the emergency care provided. For example, the resuscitation team was blamed for failed resuscitations instead of being appreciated and recognized for successful resuscitations in the EDs. The resuscitation team did not receive debriefing sessions after failed resuscitations. The resuscitation teams received different practices and instructions from various team leaders about standardized procedures and policies of the resuscitation process which negatively impacted the emergency care services. The following sub-themes for optimum resuscitation processes discuss the diverse challenges experienced in detail:

#### 3.1.1. Sub-Theme: A General Shortage of Trained Staff Members for Achieving Optimum Resuscitation

The study’s medical doctors and professional nurses reported a general shortage of trained staff to render emergency care services in the EDs. Participants reflected that a lack of trained staff members results in a lack of direction and no understanding of one’s role in the EDs, which impacts the provision of optimum resuscitation. The following extracts support the findings:

**Professional Nurse 4:** 
*“Due to shortage of trained staff, you find that the resuscitation room is not in order and people don’t know what they are doing, they don’t know their roles during resuscitation which affects optimum resuscitation”.*


**Professional Nurse 2:** 
*“The challenge we encounter is that we don’t have more trauma-trained nurses and some doctors are not conversant with resuscitation”.*


**Medical Doctor 5:** 
*“There are not enough trained nurses and doctors”.*


#### 3.1.2. Sub-Theme: The Lack of Emergency Equipment and Poorly Maintained Equipment for a Resuscitation Process

Emergency medical equipment is a critical component of the resuscitation process in EDs. Participants reported a shortage of emergency equipment for intubation and diagnostic purposes, affecting the optimum resuscitation process. The following quotes support the participants’ responses:

**Professional Nurse 5:** 
*“Equipment that is not working frustrates us during resuscitation, you find that you send someone to request that equipment during the resuscitation process and the time the equipment is available, the patient has complicated, that is one reason why we lose patients”.*


**Medical Doctor 1:** 
*“Lack of equipment, faulty equipment, is the main one on my list, as poor training, and staff development. A simple thing like a blood gas machine for instance now we don’t have, an ultrasound machine which is a basic examination tool, it is not even a special examination tool. It’s like a stethoscope, It’s like a diagnostic Ear, Nose, and throat set, that thing must be by the patient’s bedside, we don’t have such”.*


**Medical Doctor 4:** added by saying, *“I mean, for blood gas, now, I just did a blood gas, I walked for almost about a kilometer, I had to leave casualty where I should be attending patients because I’m chasing the machine which is probably the only one working in the complex. Basic things…we have got this laryngoscope that is not working well due to factory fault, you try to intubate, the light will turn off inside patient’s mouth”.*

**Medical Doctor 2:** 
*“Lack of equipment also plays a major role. Eh, for example, you want to do an x-ray quickly on the patient, there is only one mobile machine may be in the hospital. The time you call for that x-ray, you find that the radiographer went somewhere with that machine, so you have to wait, some of the equipment will be the blood gas machine”.*


#### 3.1.3. Sub-Theme: Resuscitation Teams Receive Conflicting Instructions from Various Team Leaders about Emergency Care

The participants indicated that they experience challenges with contradicting instructions and the lack of standardized procedures in the EDs. The instructions given by the resuscitation team leader often contradict the desire of the medical doctors. The following quotes support the sub-theme:

**Professional Nurse 1:** 
*“Usually, the nurse will observe that patient is deteriorating and needs intubation, suggest to the doctor, then the doctor disagrees with a team leader about the need for intubation…in most instances ignore the nurse, but you find that by the time the doctor realizes that the patient indeed needed intubation, the patient has already complicated”.*


**Professional Nurse 2:** 
*“We also need a standardized method to guide us…you will find different people, eh, maybe doctors, giving different orders, maybe even arguing in front of patients, one saying I’ve been trained, we are doing things like this and the other one saying I’ve also trained at such and such a place and were doing things like this…”*


**Professional Nurse 9:** 
*“During resuscitation, you find that maybe the patient was in room one and you notice that he/she is not doing well and move him/her to the resuscitation room for assistance, but the team leader will inform you to return the patient to the main cubicle…”*


#### 3.1.4. Sub-Theme: The Lack of Counselling and Debriefing as a Support System by Management and Supervisors

Participants indicated that the management and supervisors do not appreciate and recognize their good work of resuscitating the patients in the ED. Instead, the resuscitation team is always blamed for the failed process, thus affecting their morale. The following extracts support this:

**Medical Doctor 5:** 
*“I don’t even know the role of management in the ED because what they know is to blame us. But as for us, we usually sit on morbidity and mortality meetings, discuss the death and teach one another from the pitfalls identified”.*


**Professional Nurse 2:** 
*“No, no support, we don’t have support, they only see the mistake that you are doing, but when you do good things they don’t see you. They don’t even appreciate…It hurts us because somebody cannot die and we continue like nothing happened. That is why I say we don’t get any support”.*


**Professional Nurse 12:** 
*“… If we can have a disaster and inform them timeously they don’t come, they will only come later when the disaster has been called off. What they will ask is the statistics, ‘How many priorities one, two or three did you have? How many death did you have?’ and then they’ll leave. They won’t even appreciate the excellent work you’re doing”.*


**Medical Doctor 3:** 
*“Post resuscitation, there is no support, you just reassure yourself, you say maybe I should have done this, maybe I should have done that, maybe if the surgeon had come in early, there is no support, it’s like you’re counselling yourself”.*


**Professional Nurse 10:** 
*“We use a platform of our mortality meetings to discuss resuscitated patients. Especially files where things went wrong, and sit down and discuss them…Its wrong, we need a debriefing session”.*


#### 3.1.5. Sub-Theme: The Existence of Trauma Related to Failed Resuscitation Efforts

The current study reported the resuscitation team are subjected to stress and trauma, post-resuscitation, especially if it is a failed resuscitation. The resuscitation team expressed concern regarding the lack of trauma counsellors for emotional support after a failed resuscitation activity in the EDs. The following quotes support the findings:

**Professional nurse 2:** 
*“When I started working here some few years back, I found that they were having trauma counsellors so that after treating those patients, we were counselled, whether the patient survives or not, we were supposed to have counselling after treating patients. These days we don’t have trauma counsellors, we don’t have debriefing sessions, we don’t have anything, they will give a report that we have succeeded in resuscitating the patient so and so…”*


**Professional nurse 3:** 
*“…There is no psychologist arranged to counsel us, we just live in another environment, we see these people dying and nothing is done to us. We are not being taken care of as staff…”*


**Medical Doctor 3:** 
*“No, you become discouraged until you go home; no one will give you that support. There’s no room for post-counselling or anything. Once the resuscitation becomes unsuccessful, you are on your own… You have to recover first because there’s no support after resuscitation, whether successful or not. There’s no structure for support, and even after the patient’s death there is no post-counselling, only families will be counselled, but as for the doctor, you are on your own…”*


**Medical Doctor 2:** 
*“We don’t get counselling sessions, the only time we got emotional support was a long time ago when there was an accident where a Condor which was supposed to carry ten passengers carried sixteen passengers and got involved in a motor vehicle accident”.*


#### 3.1.6. Sub-Theme: The Lack of Available Funds for Payment of a Continuous Training Program Causes Staff to Feel Left behind with Emergency Skills

All participants expressed concern about the lack of continuous training to update their skills and knowledge about emergency care. They reported that there was previously free Basic Life Support training, which is no longer available due to budget constraints. The following extracts support the findings:

**Professional Nurse 3:** 
*“Previously, we received Basic Life Support training, but now they say we must pay for ourselves to be trained…”*


**Medical Doctor 1:** 
*“We don’t have in-service training, apparently because of budget issues, it is only when you have money then you’ll do it privately”.*


**Medical Doctor 2:** 
*“I also think lack of training of the team members, there was a time when these courses were offered for free,… at the moment now you have to use your own money. They are not even offered in our province”.*


**Medical Doctor 4:** 
*“Our Head of Department does encourage us to do emergency management courses but the problem might be finances for most people because these courses are expensive”.*


## 4. Discussion of Results

The study aimed to describe the challenges for optimum resuscitation processes as experienced by the resuscitation team in the EDs of Limpopo Province. The study’s findings identified a shortage of human and material resources in both EDs of the selected hospitals, which impacts the provision of optimum resuscitation. The shortage of trained staff in all categories (professional nurses and medical doctors) is one of the major challenges experienced in EDs. The study results revealed that though EDs are highly demanding, the workload for the resuscitation team remains challenging, especially with a limited number of trained and untrained medical doctors and professional nurses. Globally, it is documented that EDs are stressful environments for staff members, with increasing numbers of casualties presenting resulting in high pressure and high volume workloads [8,9,11]. In SA, the National Education, Health and Allied Workers Union (NEHAWU) has criticized the Eastern Cape Department of Health over conditions in the Emergency Unit of Livingstone Hospital, Port Elizabeth. The association further raised that at the beginning of 2018, the emergency clinic was confronted with a shortage of staff, with a portion of them at retirement age, and this was exacerbated by some being deceased [26]. The above studies supported the need for adequate resuscitation team staffing to cater to an increasing number of emergency casualties and close the gap in the staff complement of EDs. Creating awareness and addressing these challenges will improve the quality of emergency health care and reduce medico-legal hazards associated with the shortage of trained emergency personnel.

A major challenge that the resuscitation team experienced is the shortage of emergency equipment and poor maintenance. The resuscitation team reported that they struggle to obtain simple equipment such as a blood gas machine, stethoscope, and diagnostic ear, nose, and throat sets, which are supposed to be at the patient’s bedside during emergency and non-emergency situations. The fact that the two selected hospitals are tertiary academic hospitals, with a high intake and overflow of patients from the five districts in Limpopo province exacerbates the situation. The shortage and malfunctioning of medical equipment impact the provision of optimum resuscitation since the equipment is essential in the prevention, diagnosis, and treatment of disease and for the rehabilitation of patients. The challenge of shortage and malfunctioning of medical equipment has been identified in low-and middle-income countries [27]. In developing countries, the World Health Organization (WHO) estimates that 50 to 80 per cent of medical equipment is not functioning well which forms a barrier to healthcare delivery [28].

The resuscitation team raised concerns about conflicting instructions from team leaders and medical doctors in the EDs of the two selected hospitals. These findings indicate the need to implement standardized procedures and guidelines while providing emergency care. For example, a study by Vafaei, Akhtari, Heidari, and Hosseini [29] has identified the importance of having resuscitation guidelines in EDs because they provide a logical, sequential algorithmic approach. Our findings provide the baseline for developing procedures and guidelines that will support the resuscitation teams.

The resuscitation team is blamed for failed resuscitations which are overshadowed by depreciation and disregard for successful resuscitation by team leaders and managers. Our study findings indicate that the resuscitation team is often not appreciated for successful resuscitation activity. However, they are always blamed for a failed resuscitation activity which subjects them to psychological problems. Our findings indicate the need for effective debriefing and supporting the psychological aspects of the resuscitation teams. Spencer, Nolan, Osborn, and Georgiou [30] agreed with the study findings when they highlighted that the healthcare staff’s well-being and burnout are of significant concern with implications for staff attrition rates and, in turn, patient care, satisfaction, and safety. Due to this, ED nurses have higher rates of absenteeism and sick leave, decreased work performance, more work-home conflicts, and more intentions to leave the profession than nurses who work in other environments. These effects relate to ED nurses’ considerable psychological job demands and a perceived lack of supervisor support. Yuwanich, Akhavan, Nantsupawat and Martin [31] discovered that during data collection, one senior nurse described having to perform a physician’s tasks while waiting for the physician to arrive at the ED and deemed these circumstances as an untenable situation to work in.

It was discovered in the current study that resuscitation teams of both EDs from the selected hospitals do not receive support from the management and supervisors. One of the participants further outlined that management does not come to support them, even during a disaster. They will only come to gather statistics and cross-question them when things go wrong without appreciating what they did right.

The resuscitation team described challenges concerning the lack of funds for payment of continuous training programs as leaving them behind with emergency skills. Similarly, McKay, Walker, Brett et al. [32] outline the importance of healthcare sectors beginning to focus on training team-working skills as one way of reducing the rate of adverse events. Training team-working skills integrate evidence-based content and practising lifesaving skills in individual and team-based clinical environments [33]. This study further revealed that participants were told that there were no funds when they were supposed to attend national and international trauma short courses. They became frustrated when they had to pay for themselves to acquire skills that promote patient survival rates. Our findings suggest lack of support for training and staff development by the hospital management following the skills development Act 97 of 1998 in South Africa.

## 5. Limitation of the Study

The study took place at the emergency units of the two tertiary hospitals in the Capricorn District, Limpopo Province, South Africa. The findings of this study cannot be generalized to other settings. The study was conducted during the COVID-19 pandemic; therefore, the availability and access to the participants were a challenge.

## 6. Conclusions

This study’s results indicated that the resuscitation teams of the two identified tertiary hospitals face diverse challenges that cause them stress and burnout. The challenges result from a lack of trained and experienced personnel and material resources. The trauma counsellors and management must provide psychological support, debriefing or counselling post-failed resuscitation for optimum work performance. The study recommends a sustainable appreciation and debriefing programme in the emergency unit as a support system for the emergency personnel staff. We recommend future studies for addressing the identified challenges as a way to provide support to the resuscitation teams in the EDs.

## Data Availability

Data are not shared due to privacy and ethical restrictions.

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
