# Peer review of "Challenges for Optimum Cardiopulmonary Resuscitation in the Emergency Departments of Limpopo Province: A Qualitative Study"

_healthcare, 2023, doi:10.3390/healthcare11020158_

Round 1

Reviewer 1 Report

Dear authors,

I have read your manuscript with interest, which brings to light the difficult work of doctors and nurses in low-resource settings.

However, there are several critical considerations I want to share with you. For convenience, I will follow the structure of the article itself.

Introduction

Paragraphs in lines 56 and 60 repeat the same concepts by different cited authors. Aside from the repetition, it seems these references are not consistent with what is supported by Muthelo et al.

In fact, the two articles [6 and 9] deviate from the very subject of the manuscript as follows:
- article [6] cover specifically the topic of triage knowledge and skills, with no mention of the effects on the resuscitation team nor about how stressful the ER environment is;
- article [9] never specifically mention patients with “cardiovascular-related conditions” nor their increasing numbers. Again, there is no mention of the “resuscitation team”, as the study [9] targets a sample of Australian nurses and doctors working in two different Australian Emergency Departments, never referring to resuscitation dynamics.

Line 79 mentions a study in which “most of cardiac arrest patients were dissatisfied with the waiting period in the EDs”: after reading the cited article, it seems this statement cannot be substantiated.

Methodology

Please review punctuation in lines 100-102

Results

Review numbering in table 3.
It would be good to clarify which profile fits the team leader: is it the head of the department (doctor or nurse?), or is it a functional position that rotates among the clinical staff?

Discussion

Line 315: please correct the reference as the original source is a WHO document of 2012.
Line 347-8: “A previous study was done by [28] agree…” - aside from missing the authors, it appears again there is again an incongruous citation, as the study was measuring PTSD and debriefing practices. It is not clear where it refers to a lack of leadership and poor team dynamics within the original article.

It is understood that the authors intend for “resuscitation team” members any healthcare professional working in the ED, but this could lead to some confusion.  Would be possible to clarify this since the beginning of the article?  

In my view, the article needs a thorough review before publishing.

Thank you

Author Response

TABLE OF CORRECTIONS COMMENTS FROM REVIEWER 1

COMMENT

CORRECTIONS

Introduction:

-          Article [6 and 9] deviate from the very subject of the manuscript

Introduction

Line 52-58 were deleted as the information provided is not in line with the contents of articles 6 and 7. Contents of articles 6 and 7 review. Article 8 and its contents were reviewed and moved to 11 where it is relevant. 

Introduction

-          Line 56-60 repeat same concepts by different authors. References are not consistent with what is supported by Muthelo et al.

-          Article 8 First author left out incorrect initials of the authors and  incorrect citing

Introduction

-          Article 8 deleted as it is irrelevant. The contents of Article 9 and 10 moved to line 59, and 66 respectively to suit what the articles are about.

-          The first author included Stehman CR, Testo Z, Gershaw RS, Kellogg AR, initials reviewed and the article cited correctly as number 11.

Introduction

Line 79 mentions a study in which “most cardiac arrest patients were dissatisfied with the waiting period in the EDs”. After reading the cited article, it seems this statement cannot be substantiated.

Introduction

Line 79 reviewed and deleted based on the contents of the article which has nothing to do with Cardiac arrest patients being dissatisfied with waiting period

Methodology

Review punctuation in line 100-102

The study was conducted in in the EDs of two tertiary hospitals situated in the tertiary Hospital A is a level three hospital with twenty-two wards and fourteen specialized clinics of the Outpatient Department. The EDs fourteen cubicles which include two resuscitation units (trauma and medical/surgical) and one triage.   

Methodology

Line 100- 102 punctuation reviewed and corrected

Results

-Review numbering in table 3

- Clarify which profile fits the team leader: is it the head of the department (doctor or nurse?) or is it a functional position that rotates among the clinical staff?

Results

Numbering reviewed and corrected

Discussion

Line 315:please correct the reference as the original source is a H

WHO document of 2012

Discussion

Corrected line 229 and reference 28

Discussion

Line 347-348: “A previous study was done by [28] agree…”aside from missing the authors it appears again there is an incongruous citation as the study was measuring PTSD and debriefing practices. It is not clear where it refers to a lack of leadership and poor team dynamics within the original article 

Discussion

Line 347 to 351 was deleted because article [28] does not deal with leadership but deals with staff well-being and burnout. The authors were corrected earlier on to read  Spencer,  Nolan, Osborn, and Georgious [27]

Discussion

Clarify who the members of the “resuscitation team” are does it include any member of healthcare professional working in the ED? Clarify this in the article to prevent confusion.      

Discussion

Discussion

Thorough review of the article

Discussion

All the reviewer’s comments were reviewed.

Reviewer 2 Report

The authors discussed a variety of obstacles to optimal resuscitation procedures in emergency departments. Followed by questions and comments.

1. The format is improper; please refer to the standard format of journal. kindly correct it.

2. Please clarify the parameters or create a table in this manuscript.

3. The purpose of this thesis is strong and useful. The motivation will be tougher if the comparison between literatures has given.

4. In the Introduction Section, please provide a complete flowchart of current state of the research field. Key publications should be cited as completed as possible. Please also clarify the novelty and application implication of your work in this Section.

5. In Discussion Section: Please compare your results with data published previously and provide a clear interpretation of your results.

6. The format in Table 3 Sub-Themes 1.6 is incorrect; please correct it.

In summary, the article is sound and acceptable if the authors do revise the manuscript based on comments as above-mentioned.

Author Response

Reviewer 2 Comments

Comments

Corrections

In the Introduction Section, please provide a complete flowchart of the current state of the research field. Key publications should be cited as completed as possible. Please also clarify the novelty and application implication of your work in this Section.

The last paragraph of the introduction indicate the novelty and application of the study. We also added relevant and recent literature (key publications) when correcting the flow of the introduction.

 In the Discussion Section: Please compare your results with data published previously and provide a clear interpretation of your results.

The statements from 339 to 347 compare the study findings to previous studies. There comparison is followed by authors interpretation.

The format in Table 3 Sub-Themes 1.6 is incorrect; please correct it.

The format has been corrected as suggested

The format is improper; please refer to the standard format of journal. kindly correct it

The format has been effected as suggested. We added the headings according to the journal requirements such as conflict of interest, authors contribution and funding.

Reviewer 3 Report

The manuscript describe the challenges face by resuscitation teams in emergency departments from two hospitals in Africa. The authors collected data through semi-structured interviews which were analyzed using thematic analysis.

There are some aspects you should improve:

- the introduction section is very informative however more recent references should be included.

In section 2.2:

- I would like to know the generals of the participants (gender, years of experience, age, etc.) and their affiliation (hospital A or B)

- how did you determine "the best suitable participant", did you use a test or how did you measure the level of knowledge regarding resuscitation?

- About data analysis, how did you tackle the disadvantages of thematic analysis? the themes and sub-themes were agreed with all the analyzers or were established since the beginning? More details about the analysis process should be included. 

Some minor comments:

- review section 2.1 specifically lines 103 and 104 where the writing is not clear.

- some references appear in read, why?

- the caption in table 3 as well as enumeration of sub-themes should be revised

Author Response

TABLE OF CORRECTIONS FROM REVIEWER 3

COMMENTS

CORRECTIONS

In section 2.2:

- I would like to know the generals of the participants (gender, years of experience, age, etc.) and their affiliation (hospital A or B)

A table (table 2.1) in section 2.2 with the  demographic details /generals of the participants was included however, analysis was not done in isolation per hospital A nor B hence on the table affiliation per hospital is not indicated

- how did you determine "the best suitable participant", did you use a test or how did you measure the level of knowledge regarding resuscitation?

The exclusion criteria for the study were indicated in section 2.2 under population and sampling. The study excluded all community service nurses and medical interns as these categories are still learning and working under the supervision and are presumed to be less experienced with resuscitation in the ED.

- About data analysis, how did you tackle the disadvantages of thematic analysis? the themes and sub-themes were agreed with all the analyzers or were established since the beginning? More details about the analysis process should be included

Three statements has been added regarding the concern of thematic disadvantages- authors described how the theme and sub-themes were generated as well as the measures implemented to address any inconsistencies. More details about the analysis are provided.

- review section 2.1 specifically lines 103 and 104 where the writing is not clear.

Lines 103 and 104 rephrased to improve meaning

- some references appear in read, why?

All references verified and appear the same

- the caption in table 3 as well as enumeration of sub-themes should be revised

Table 3 enumeration corrected

Round 2

Reviewer 1 Report

Thank you for your further work on this research. The improvements are substantial and support the importance of your findings.

Please find below a few considerations that I trust you will consider.

Lines 51-52 made reference to "healthcare professionals trained and untrained": this statement needs to be better contextualised.  I believe you were referring to BLS training, yet it would be great to understand better the training you are referring to.

Lines 64-65 made reference to the article [8], where it is mentioned that  "stressful EDs were related to the increasing numbers of patients with cardiovascular-related conditions": I cannot find a reference to such in the original article. Hence, I am suggesting reviewing the cited article as can provide different hints to substantiate your claims.

Author Response

REVIEWER 1 COMMENTS

Comments

Corrections

Lines 51-52 made reference to "healthcare professionals trained and untrained": this statement needs to be better contextualised.  I believe you were referring to BLS training, yet it would be great to understand better the training you are referring to.

The sentence has been rephrased as South African healthcare professionals trained and untrained in BLS.....as suggested by the reviewer.

Lines 64-65 made reference to the article [8], where it is mentioned that  "stressful EDs were related to the increasing numbers of patients with cardiovascular-related conditions": I cannot find a reference to such in the original article. Hence, I am suggesting reviewing the cited article as can provide different hints to substantiate your claims.

We reviewed the original article which guided us to rephrase the sentence according to the content in the original article

Reviewer 2 Report

Except the incorrect format, the current revision is sound and acceptable for publication.

Author Response

Dear Editor

Please see the table of corrections according to the reviewer`s comments for the manuscript.

REVIEWER 2 COMMENTS

Comments

Corrections

Except for the incorrect format, the current revision is sound and acceptable for publication

We have arranged the manuscript according to the author’s guidelines for the journal. Although the journal has no specified format of items, we have aligned the headings according to the latest article published in the journal.

Reviewer 3 Report

The authors have addressed all the previous observations made. However I still have some concerns about the population and sampling. It seams to me that the sampling method was very subjective because the participants were not evaluated as "best suitable" or the authors didn't describe that process and I think it is very important. Are the results influenced by this "best suitable" metric???

Author Response

Dear Editor

Please see the table of corrections according to the reviewer`s comments for the manuscript.

REVIEWER 3 COMMENTS

Comments

Corrections

The authors have addressed all the previous observations made. However, I still have some concerns about the population and sampling. It seems to me that the sampling method was very subjective because the participants were not evaluated as "best suitable" or the authors didn't describe that process and I think it is very important. Are the results influenced by this "best suitable" metric??

We rephrased and added a statement that indicated how the participants were suitable for participation in the study. The suitability was related to their qualification, knowledge and experience regarding working in the Eds.
